# Coverage of suicide deaths in newspapers and perceptions of suicide loss survivors on reporting: Insights from a community survey in India

Neha Dhole[1], Md. Akbar[1], Moutushi Majumder[1], Siva Prasad Dora[1], G. Anil Kumar[1], Rakhi Dandona[1,2,3]*

**1** Injury Prevention Research Centre, Public Health Foundation of India, New Delhi, India, **2** Institute for Health Metrics and Evaluation, University of Washington, Seattle, Washington, United States of America, **3** Centre for Mental Health and Community Wellbeing, Melbourne School of Population and Global Health, The University of Melbourne, Melbourne, Australia

* rakhi.dandona@phfi.org

## Abstract

Limited research explores newspaper coverage of suicide deaths, and its impact on suicide loss survivors. An exploratory inquiry study examined newspaper coverage of suicide deaths and how families perceived this reporting of suicide deaths within their family in India. Semi-structured interviews were conducted with suicide loss survivors, an adult member most knowledgeable about the suicide, for 155 suicide deaths in three Indian states – Maharashtra, Tamil Nadu and Uttar Pradesh. Newspaper coverage by socio-demography of the deceased, and support and opposition by the survivors for the newspaper reporting and their reasons for it are reported. Among all suicide deaths, 39.4% (95% CI: 31.9–47.3%) were reported in newspaper for both sexes combined, 37.8% for males and 43.2% for females, which varied by socio-demography of the deceased. Deaths registered with the police were 4.5 times more likely to be reported. For the 61 cases reported in newspaper, 49.2% survivors opposed the coverage, 31.1% supported it, and 19.7% declined to respond. The support for reporting of male suicide deaths was significantly associated with increasing levels of education of the survivor. No specific pattern was seen for female suicide deaths for supporting or opposing the reporting. Support was mainly to raise awareness about suicide (73.4%), including general awareness about suicide, implications of death, and about farmers' suicides. Majority of the survivors opposing the coverage did not provide a specific reason for it. Implications of reading about the suicide death in newspaper varied by the socio-demography of the deceased and of the survivors, highlighting the complexity of suicide bereavement and that it needs to be understood further within the local socio-cultural context. These insights from families can inform refinements to existing media codes, ensuring that they address the

**Data availability statement:** The data have sensitive detailed information on suicide and hence cannot be publicly shared. Request for data can be sent to the Institutional Review Board (trc-iec@phfi.org).

**Funding:** This work was supported by the Mariwala Health Initiative, Mumbai, India (grant number LN1201 to RD). The funder had no role in study design, data collection and analysis, decision to publish, or preparation of the manuscript.

**Competing interests:** The authors have declared that no competing interests exist.

needs of survivors, promote greater sensitivity in coverage, and equip reporters with practical approaches to minimize harm.

## Introduction

The World Health Organization (WHO) and the International Association for Suicide Prevention have formulated guidelines that emphasize several key themes for responsible media reporting of suicide [1]. These include avoiding sensationalism or graphic descriptions of the death, refraining from providing explicit details about the method, and using non-stigmatizing language that respects the dignity of the deceased and their families. The guidelines also encourage including information on help-seeking and support services, highlighting stories of hope and recovery, and promoting awareness of suicide prevention strategies. Overall, the focus is on minimizing potential harm, reducing the risk of contagion, and supporting public understanding of suicide as a preventable health issue. These guidelines are important as media reporting of suicide deaths can have both a consequential negative effect on suicide deaths and suicidal behavior among the public, [2–5] and potentially protective effect through details of how and where someone could seek help for suicidal ideation or behavior, messages of hope and alternatives to suicide, and educating the public at large about suicide [4,6].

The Press Council of India adapted the WHO guidelines in 2019 on suicide reporting by the media [7]. Much of the available literature globally, including from India, has focused on the quality of media reporting of suicide deaths [6,8–17]. There remains a striking gap in understanding how such reporting affects the bereaved families themselves. Families may experience media coverage as intrusive, stigmatizing, or retraumatizing, while others may find value in raising awareness [18,19]. These responses have important implications, not only for the wellbeing of suicide survivors but also for shaping ethical and sensitive approaches to media practice. Yet, systematic evidence on family perspectives remains limited, highlighting the need for focused inquiry in this area [20]. Notably, the survivors may be at a greater risk of suicide themselves after the loss, which is why the WHO guidelines recommend that journalists exercise caution when interviewing bereaved family members, friends, and relatives [1]. In this background, our study documented the coverage of suicide death reporting in the newspaper and the perspectives of suicide loss survivors across three states in India.

## Methodology

### Ethics statement

This study was approved by the Institutional Ethics Committee of the Public Health Foundation of India. All participants provided written informed consent, and for those who could not read or write, the participant information sheet and consent form were explained by a trained interviewer and a thumb impression was obtained. The participants were informed that a decision not to take part in the research will

have no impact on them, and that they could stop the interview at any time or decide not to answer specific questions. All participants, irrespective of the voiced need for mental health support, were provided with a list of mental health support services available closer to their homes for in-person consultation, and were also provided with the phone numbers of telephone helplines available in the state or country-wide. The research staff had the necessary skills to ask questions sensitively, respond and provide appropriate immediate support when participants displayed any sign of distress. A protocol was in place for consultation if a participant was identified as needing immediate care.

## Study design and data collection

Deaths of all ages that occurred between 2019 and 2022 were listed in a nationally representative survey of deaths covering 1 million population in nine Indian states – Assam, Gujarat, Haryana, Jharkhand, Kerala, Maharashtra, Odisha, Tamil Nadu, and Uttar Pradesh – as per the sampling criteria described elsewhere [21]. The sample size for deaths was estimated 25,800 deaths for a 1,000,000 population sample based on the crude death rate for each of the sampled state. Following the state selection, a multi-stage sampling procedure was used to select 1,000 clusters from 50 districts in the nine sampled states with the aim of having a sample representative of the population of India. The number of districts were distributed across the 9 sampled states based on the total population in these states. Four districts each were sampled in the states of Odisha, Kerala, Jharkhand, Assam and Haryana; 6 districts each in Gujarat and Tamil Nadu; 8 districts in Maharashtra; and 10 districts in Uttar Pradesh. The districts in each state were sampled randomly until a reasonable geographic spread was achieved within the state. Based on the Census 2011 data, [22] 20 study clusters of about 1,000 population each were sampled in each district using stratified systematic sampling technique, with the proportion of rural-urban clusters similar to the proportion of rural-urban population in each district. The rural-urban population proportion in each district during the study period was estimated by applying the percentage increase in urban population in each sampled state between Census 2001 and Census 2011 [22,23] to the period after 2011. In the selected districts, we divided large villages into segments of 1,000 population each, and combined villages with smaller population with others to make a cluster size of 1,000 population each. We then systematically selected the rural clusters from the villages with the first cluster sampled randomly, and similarly the urban clusters from the wards. All households in the sampled cluster were mapped and listed in a serpentine pattern starting from the north-east corner and ending in the opposite corner. A household was defined as people eating from the same kitchen. All households with at least one death between January 2019 and December 2022 were considered eligible for detailed survey. Data collection for this survey was undertaken from 17/01/2023–17/08/2023.

A total of 366 suicide deaths were listed in this national survey, and 220 (60.1%) of these were in three of the nine states - Uttar Pradesh, Tamil Nadu, and Maharashtra. These three states represent varied socio-demography in India, with Maharashtra and Tamil Nadu reflecting stronger health systems relative to Uttar Pradesh [24]. We undertook semi-structured interviews between 21/10/2024–06/01/2025 in these three states aimed at understanding the bereavement needs in the affected households, the perceptions of these households about reporting of suicide deaths in newspapers as is common in India, and reporting of suicide death to the police. The respondent for this study was the adult member who was deemed most knowledgeable about the suicide and related issues as identified by the households of the deceased. Trained interviewers collected data on the deceased's socio-demographic details, and asked if suicide death was reported in newspaper or other media. For those who confirmed the newspaper/media coverage, suicide loss survivors were asked how they felt upon reading or hearing the news, and whether they would have preferred the death not be covered in the media. They were then asked if they supported or not supported the reporting of suicide death in the media, and their reason for the same. The open-ended answers to these questions and the verbatim narratives captured by the interviewers provided the qualitative data analyzed in this paper. Taking cues from open coding in grounded-theory approach to qualitative analysis [25], ND first reviewed the themes derived from open-ended responses and verbatim narratives. The themes or codes were grouped together according to their similarity. Words used by respondents were

PLOS Mental Health

included as themes in some cases. These themes were tabled and discussed with RD and GAK, who also reviewed them and decided if the themes were appropriate. Disagreements were resolved through discussions among ND, RD, and GAK. Thus, we implemented inductive coding of the responses to explain why suicide loss survivors supported or opposed the newspaper reporting [26].

The interview questionnaire was developed in English and then translated into local languages (Hindi, Tamil and Marathi), and then again translated back to English to ensure the accuracy and relevant meaning of the questions, without diverging from the intent of the questions. To test the logistics and the quality of data collection, pilot testing of the questionnaire in Hindi was carried out on three suicide deaths in Haryana state, and on two suicide deaths each in Tamil and Marathi languages in Tamil Nadu and Maharashtra states, respectively. Appropriate modifications were made to the questionnaire in each language as required before the start of the survey in each state.

## Statistical analysis

We report on the coverage of newspaper reporting of suicide deaths by state and demographic characteristics of the deceased. We present the distribution of suicide loss survivors' opinions on newspaper reporting based on their socio-demography. In addition, we provide direct quotes from suicide loss survivors that illustrate their reasoning for said opinion/preference according to the themes we discovered. These quotes were selected on the basis of uniqueness of reasons and avoidance of repetition. We aimed to add quotes from each state, and for reporting on both deceased men and women. However, the details provided by respondents (or lack thereof) limited us to include only certain respondents' opinions on newspaper coverage. Wealth index quartiles utilized in the analysis were calculated using the standard methods used in the National Family Health Survey. [27] We report 95% confidence interval (CI) for all estimates, and chi-square and z test of significance as relevant. All analyses were performed using STATA V.13.1 software (StataCorp, USA).

## Results

### Participation

Detailed in-depth interviews were available for 155 (70.5% participation rate) of 220 suicide deaths, including 111 (71.6%) male suicide deaths and 120 (77.4%) suicide deaths in rural areas.

### Newspaper coverage of suicide death

Newspaper and/or other media coverage of suicide death was reported for 62 deaths, of which 61 (98.4%) were reported in a newspaper. The newspaper coverage of suicide deaths was estimated at 39.4% (95% CI: 31.9-47.3), 37.8% (95% CI: 29.2-47.3), and 43.2% (95% CI: 29.1-58.4) for both sexes combined, male and female suicide deaths, respectively. For both sexes combined, Tamil Nadu state had the lowest newspaper coverage at 28.8% (95% CI: 18.1-42.6) compared to Uttar Pradesh (47.2%; 95% CI: 31.6-63.4) and Maharashtra (43.3%; 95% CI: 31.9-55.4).

Table 1 documents the newspaper coverage of suicide deaths by a variety of socio-demographic characteristics of the deceased. The states of Maharashtra and Uttar Pradesh had the highest newspaper coverage for male (44.1%; 95% CI: 31.9-57.0) and female (52.6%; 95% CI: 30.5-73.7) suicide deaths, respectively. More newspaper coverage was noted for urban suicide deaths for both males (44.0%; 95% CI: 26.1-63.6) and females (60.0%; 95% CI: 29.0-84.6) compared to rural deaths, but the differences were not statistically significant. The coverage was relatively higher if the deceased was either salaried or a farmer for both male and female suicide deaths.

The highest newspaper coverage among female suicide deaths was recorded for those never married (50.0%; 95% CI: 23.8-76.2), between the ages of 10–19 years (66.7%; 95% CI: 25.9-92.0), and for those belonging to wealth index quartile 2 (57.1%; 95% CI: 22.2-86.2). This coverage among male suicide deaths was the lowest for the age group 10–19 years

**Table 1. Newspaper coverage of suicide death by select socio-demography of the deceased, by sex.**

| Socio-demography of the deceased | Variable category | Male suicide deaths | | Female suicide deaths | |
|---|---|---|---|---|---|
| | | Number | Coverage of newspaper reporting of suicide death (% of variable category) | Number | Coverage of newspaper reporting of suicide death (% of variable category) |
| Overall | | 111 | 37.8 (29.2-47.3) | 44 | 43.2 (29.1-58.4) |
| Age group (years) | 10-19 | 8 | 12.5 (1.7-54.3) | 6 | 66.7 (25.9-92.0) |
| | 20-29 | 30 | 40.0 (24.2-58.3) | 22 | 36.4 (18.9-58.3) |
| | 30-39 | 20 | 45.0 (25.1-66.6) | 8 | 50.0 (19.4-80.6) |
| | 40-49 | 19 | 31.6 (14.8-55.1) | 5 | 40.0 (9.6-80.8) |
| | 50 or more | 34 | 41.2 (26.0-58.3) | 3 | 33.3 (4.1-85.5) |
| Marital status | Never married | 32 | 34.4 (20.0-52.3) | 12 | 50.0 (23.8-76.2) |
| | Ever married | 79 | 39.2 (29.0-50.5) | 32 | 40.6 (24.9-58.6) |
| Wealth index quartile | Quartile 1 | 25 | 36.0 (19.8-56.2) | 14 | 35.7 (15.3-63.1) |
| | Quartile 2 | 28 | 28.6 (14.9-47.8) | 7 | 57.1 (22.2-86.2) |
| | Quartile 3 | 33 | 36.4 (21.8-53.9) | 13 | 38.5 (16.5-66.4) |
| | Quartile 4 | 25 | 52.0 (32.9-70.5) | 10 | 50.0 (21.8-78.2) |
| Occupation | Home duties | 9 | 33.3 (11.0-67.0) | 23 | 39.1 (21.4-60.3) |
| | Labourer | 36 | 16.7 (7.6-32.7) | 4 | 0 |
| | Farmer | 24 | 54.2 (34.4-72.7) | 4 | 50.0 (11.7-88.3) |
| | Salaried | 18 | 61.1 (37.6-80.4) | 3 | 66.7 (14.5-95.9) |
| | Business/ self-employed | 12 | 33.3 (12.9-62.7) | 0 | |
| | Student | 12 | 41.7 (18.3-69.5) | 10 | 60.0 (29.0-84.6) |
| Urbanicity | Urban | 25 | 44.0 (26.1-63.6) | 10 | 60.0 (29.0-84.6) |
| | Rural | 86 | 36.0 (26.5-46.8) | 34 | 38.2 (23.3-55.8) |
| State | Maharashtra | 59 | 44.1 (31.9-57.0) | 8 | 37.5 (12.1-72.4) |
| | Tamil Nadu | 35 | 25.7 (13.9-42.7) | 17 | 35.3 (16.4-60.3) |
| | Uttar Pradesh | 17 | 41.2 (20.9-65.0) | 19 | 52.6 (30.5-73.7) |
| Police case registered for the death | Yes | 76 | 50.0 (38.8-61.2) | 28 | 60.7 (41.1-77.4) |
| | No | 35 | 11.4 (4.2-27.5) | 16 | 12.5 (2.8-41.7) |

(12.5%; 95% CI: 1.7-54.3), was the highest for ever married (39.2%; 95% CI: 29.0-50.5), between the ages of 30–39 years (12.5%; 95% CI: 1.7-54.3), and those belonging to wealth index quartile 4 (52.0%; 95% CI: 32.9-70.5).

A total of 104 suicide deaths were registered with the police for both sexes combined. The newspaper coverage was significantly higher (4.5 times) for the suicide deaths registered with the police (55, 52.9%) compared with those not registered (6, 11.8%). Half of the male deaths (Chi-square test, p<0.001) and 60.7% of female deaths (Chi-square test, p=0.002) received newspaper coverage among the suicide deaths that were registered with the police.

## Opinions about reporting in the newspaper

For the suicide deaths covered in newspaper, the distribution of socio-demography of the suicide loss survivors and their preference for newspaper reporting is shown in Table 2. A significantly higher proportion of suicide loss survivors were males for female suicide deaths (73.7%), spouses accounted for 35.7% of suicide loss survivors for male suicide deaths and whereas parents or parents-in-law accounted for 36.8% of suicide loss survivors for female suicide deaths.

For the 61 suicide deaths reported in newspapers, 30 (49.2%; 95% CI:36.7-61.8) suicide loss survivors opposed the coverage, 19 (31.1%; 95% CI: 20.6-44.0) supported the newspaper coverage, while 12 (19.7%; 95% CI: 11.4-31.8)

**Table 2. Survivor preference for newspaper reporting of suicide deaths based on socio-demography of the respondent, for the deaths reported in newspaper.**

| Survivor characteristics | Characteristics category | Male suicide deaths reported in newspaper | | | | | Female suicide deaths reported in newspaper | | | | |
|---|---|---|---|---|---|---|---|---|---|---|---|
| | | Total N=42 (% to N) | Support reporting (% to characteristics category) | Oppose reporting (% to characteristics category) | Cannot say (% to characteristics category) | p-value for Chi-square test | Total N=19 (% to N) | Support reporting (% to characteristics category) | Oppose reporting (% to characteristics category) | Cannot say (% to characteristics category) | p-value for Chi-square test |
| Overall | | 42 | 11 (26.2) | 21 (50.0) | 10 (23.8) | | 19 | 8 (42.1) | 9 (47.4) | 2 (10.5) | |
| Sex | Male | 18 (42.9) | 5 (27.8) | 8 (44.4) | 5 (27.8) | 0.805 | 14 (73.7) | 8 (57.1) | 6 (42.9) | 0 | 0.013 |
| | Female | 24 (57.1) | 6 (25.0) | 13 (54.2) | 5 (20.8) | | 5 (26.3) | 0 | 3 (60.0) | 2 (40.0) | |
| Age group (in years) | 20-29 | 7 (16.7) | 5 (71.4) | 2 (28.6) | 0 | 0.075 | 5 (26.3) | 1 (20.0) | 3 (60.0) | 1 (20.0) | 0.832 |
| | 30-39 | 13 (31.0) | 3 (23.1) | 6 (46.2) | 4 (30.8) | | 4 (21.1) | 2 (50.0) | 1 (25.0) | 1 (25.0) | |
| | 40-49 | 13 (31.0) | 1 (7.7) | 8 (61.5) | 4 (30.8) | | 5 (26.3) | 3 (60.0) | 2 (40.0) | 0 | |
| | 50-59 | 5 (11.9) | 1 (20.0) | 4 (80.0) | 0 | | 3 (15.8) | 1 (33.3) | 2 (66.7) | 0 | |
| | 60+ | 4 (9.5) | 1 (25.0) | 1 (25.0) | 2 (50.0) | | 2 (10.5) | 1 (50.0) | 1 (50.0) | 0 | |
| Relationship with the deceased | Parents/parents-in-law | 12 (28.6) | 2 (16.7) | 7 (58.3) | 3 (25.0) | 0.818 | 7 (36.8) | 2 (28.6) | 5 (71.4) | 0 | 0.005 |
| | Spouse | 15 (35.7) | 4 (26.7) | 7 (46.7) | 4 (26.7) | | 5 (26.3) | 5 (100) | 0 | 0 | |
| | Children/Children-in-law | 10 (23.8) | 4 (40.0) | 5 (50.0) | 1 (10.0) | | 3 (15.8) | 0 | 3 (100) | 0 | |
| | Sibling/Sibling-in-law | 5 (11.9) | 1 (20.0) | 2 (40.0) | 2 (40.0) | | 4 (21.1) | 1 (25.0) | 1 (25.0) | 2 (50.0) | |
| Education status | Never been to school | 4 (9.5) | 1 (25.0) | 3 (75.0) | 0 | 0.012 | 0 | | | | 0.167 |
| | Class 1-5 | 5 (11.9) | 2 (40.0) | 2 (40.0) | 1 (20) | | 4 (21.1) | 2 (50.0) | 2 (50.0) | 0 | |
| | Class 6-10 | 18 (42.9) | 0 | 10 (55.6) | 8 (44.4) | | 10 (52.6) | 5 (50.0) | 5 (50.0) | 0 | |
| | Higher than class 10 | 15 (35.7) | 8 (53.3) | 6 (40.0) | 1 (6.7) | | 5 (26.3) | 1 (20.0) | 2 (40.0) | 2 (40.0) | |
| Wealth index quartile | Quartile 1 | 9 (21.4) | 1 (11.1) | 5 (55.6) | 3 (33.3) | 0.879 | 5 (26.3) | 2 (40.0) | 3 (60.0) | 0 | 0.329 |
| | Quartile 2 | 8 (19.1) | 3 (37.5) | 3 (37.5) | 2 (25.0) | | 4 (21.1) | 2 (50.0) | 2 (50.0) | 0 | |
| | Quartile 3 | 12 (28.6) | 3 (25.0) | 7 (58.3) | 2 (16.7) | | 5 (26.3) | 3 (60.0) | 2 (40.0) | 0 | |
| | Quartile 4 | 13 (31.0) | 6 (30.8) | 4 (46.2) | 3 (23.1) | | 5 (26.3) | 1 (20.0) | 2 (40.0) | 2 (40.0) | |
| Urbanicity | Urban | 11 (26.2) | 1 (9.1) | 7 (63.6) | 3 (27.3) | 0.318 | 6 (31.6) | 1 (16.7) | 3 (50.0) | 2 (33.3) | 0.058 |
| | Rural | 31 (73.8) | 10 (32.3) | 14 (45.2) | 7 (22.6) | | 13 (68.4) | 7 (53.9) | 6 (46.1) | 0 | |

declined to respond. Support for newspaper reporting by the socio-demography of deceased is shown in Figs 1 and 2, respectively. Support for newspaper reporting of female suicide deaths varied significantly across states (p<0.001) with all suicide loss survivors in Tamil Nadu opposed to media coverage while 70% in Uttar Pradesh supported it. Support was over three times higher in rural areas (53.9%) compared to urban areas (16.7%; Chi-square test, p=0.058). For male suicide deaths, state-wise differences in support were not statistically significant (Chi-square test, p=0.126), however no survivors in Tamil Nadu supported media coverage.

On considering the opinions on newspaper reporting by socio-demography of the suicide loss survivors (Table 2), female survivors were more likely than males to oppose the reporting for both male (54.2%, z test p=0.213) and female suicide deaths (60.0%, z test p=0.547) though this was not statistically significant. Support for the reporting of male suicide deaths was associated with increasing suicide loss survivors' level of education (Chi-square test, p=0.012). No specific pattern was noticed by socio-demography of the survivors for female suicide deaths for supporting or opposing the reporting.

Table 3 documents the reasons for the suicide loss survivors supporting or opposing the newspaper coverage along with exemplary quotes from them. Support was predominately to improve public awareness for suicide prevention (73.4%), which included general awareness about suicide, awareness of the implications of death, and about farmers' suicides. Majority of the suicide loss survivors who opposed the coverage did not provide a specific reason for their opposition (16; 53.3%). The themes of support from government, stigma, and defamation were cited both in support and in opposition to the reporting. For example, avoidance of defamation through the news report was cited as a reason to support the reporting of suicide death of a married woman. In this case, the deceased's husband said that the newspaper published the accurate reason of his wife's suicide as her extramarital affair, and this reporting prevented his family from being accused of suicide abetment (case 6, Table 3). On the other hand, a suicide loss survivor cited defamation due to inaccurate reporting in the newspaper as a reason to oppose newspaper reporting (case 14, Table 3). A total of 7 (23.3%) respondents reported feeling distressed on reading the news and hence opposed it. We also found a report of suggested

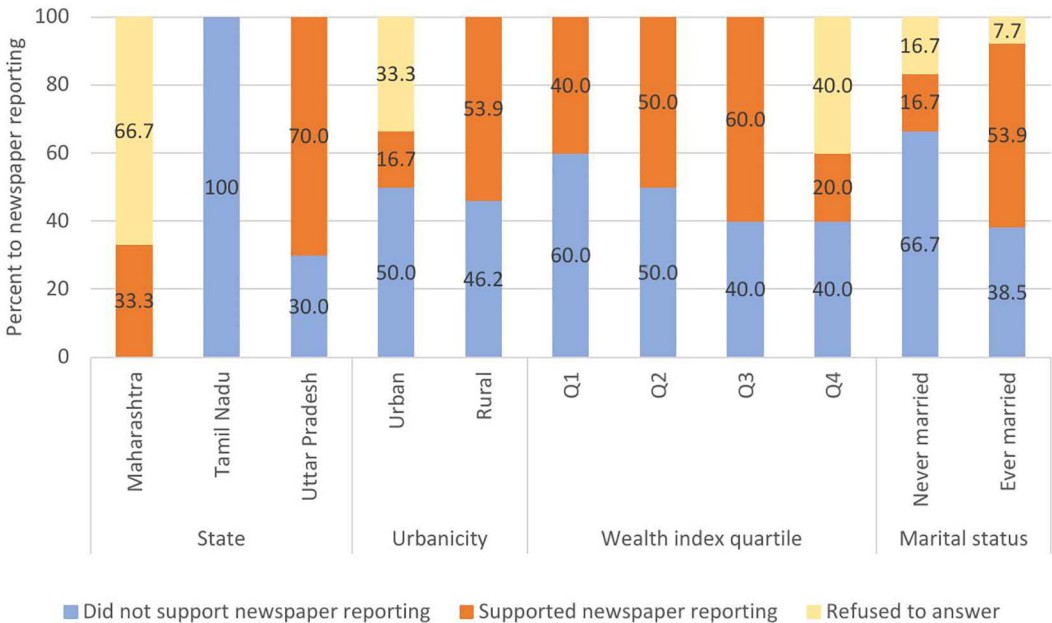

**Fig 1. Suicide loss survivor preferences for newspaper reporting of female suicide death based on socio-demography of the deceased.**

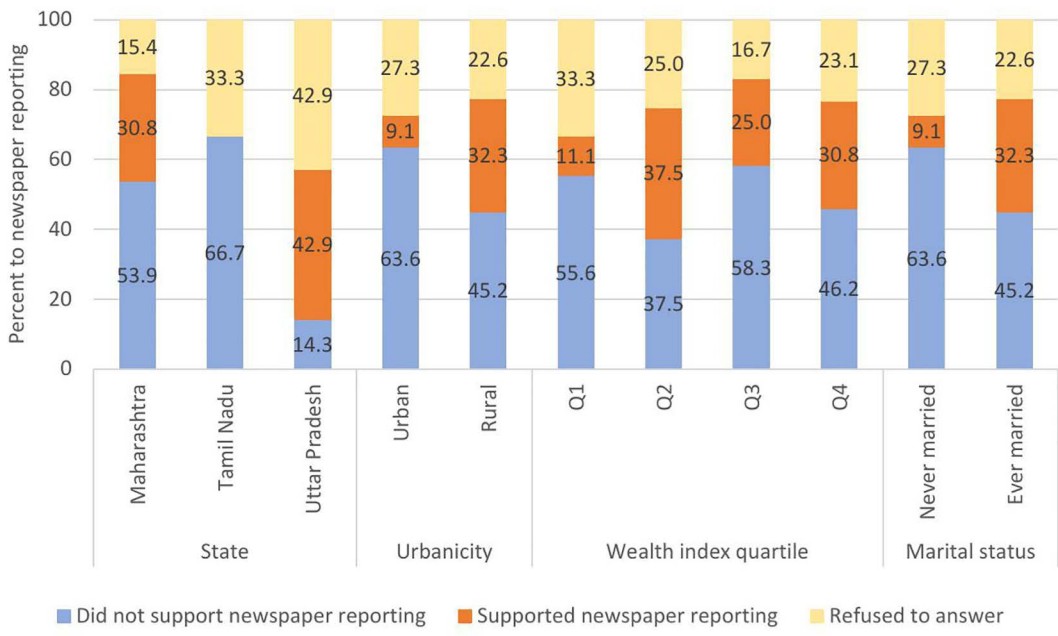

**Fig 2. Suicide loss survivor preferences for newspaper reporting of male suicide death based on socio-demography of the deceased.**

misreporting by a media person for a suicide by drowning of a 26-year-old woman in Uttar Pradesh. The media person suggested to the deceased's husband to change his narrative for the newspaper reporting from suicide death to unintentional drowning death during bathing to claim USD 5,840 under a government scheme. Two households in Uttar Pradesh showed the newspaper report to the study team, and both reports had the deceased's name with their photograph.

## Discussion

To the best of our knowledge, this is the first study to examine the coverage of reporting of suicide deaths in newspaper and the perceptions of the suicide loss survivors on this reporting. We found nearly three in seven suicide deaths in the population-representative sample of suicide deaths being reported in a newspaper. Nearly half of the suicide loss survivors opposed the newspaper reporting of suicide death, and the survivors' perceptions and support for newspaper reporting varied by state and by their own and deceased's socio-demography. These findings highlight a substantial gap in the understanding and development of postvention and suicide bereavement support in India, and have implications for media reporting on suicide deaths by taking into account the impact on suicide loss survivors.

A pattern was observed in the newspaper coverage of suicide death with a higher coverage for urban deaths, a lower coverage in Tamil Nadu state, and varied reporting by the sex of the deceased. A higher coverage of female deaths in 10–19 years age group was reported whereas a suicide death in this age group for males was reported the least. This was also reflected in reporting by marital status with reporting higher for never married females and relatively lower reporting for males based on marital status. Notably, interpretation of this pattern should take into consideration the wide confidence intervals for some findings given the small sample size. Suicide deaths of females, those under 30 years of age, females who were unmarried at the time of death, and separated or widowed males were over-reported in the media in a previous study undertaken in the Indian state of Tamil Nadu [28]. Suicide reporting is considered "newsworthy" in India and reporting on particular population groups is favored to capture readers' attention [29]. The journalists in India have

**Table 3. Reasons given by the respondents for supporting and opposing the newspaper coverage of suicide death reported as verbatim. Reasons are not mutually exclusive.**

| Reasons for supporting newspaper reporting (Age, sex, and state of the respondent) | Reasons for opposing newspaper reporting (Age, sex, and state of the respondent) |
|---|---|
| **To increase public awareness about suicide**<br>• "We wanted people to know so that people should not take such steps." (62-year-old male, Uttar Pradesh, case 1)<br>• "People should know the suicide death with reason so no one can attempt." (26 -year-old female, Maharashtra, case 2)<br>• "Yes, it should come. If a person leaves two children, then what the family goes through should be reported." (27-year-old male, Uttar Pradesh, case 3)<br>• "People should know about farmer suicides." (23-year-old male, Maharashtra, case 4) | **No reason given**<br>• "We didn't like it." (45-year-old male, Tamil Nadu, case 8)<br>• "We didn't want to publish but police published." (47-year-old female, Tamil Nadu, case 9)<br>• "It should not have come, we thought it was not good." (46-year-old female, Maharashtra, case 10)<br>• "This news should not have been printed in the newspaper." 28-year-old male, Maharashtra, case 11)<br>• "News about suicide by itself should not be published." (35-year-old male, Maharashtra, case 12) |
| **Likelihood of benefit from the government as result of the case coverage**<br>• "People know the suicide death with reason so farmer can claim the *yojana* (government support scheme)." (37-year-old female, Maharashtra, case 5) | **Because reporting does not get government help**<br>• "Every family has issues but there is no support from the government; so what is the point if this comes in the newspaper or not." (24-year-old male, Uttar Pradesh, case 13) |
| **To avoid defamation through the news report**<br>• "My wife's nature was not good. She had an affair with my brother-in-law. To avoid the defamation that we killed her, I think that it was important for the correct news should be published in the media." (33-year-old male, Uttar Pradesh, case 6) | **Defamation due to inaccurate reporting**<br>• "We were defamed intentionally by reporting that we had killed our daughter-in-law. We did nothing." (60-year-old male, Uttar Pradesh, case 14) |
| **To support reducing suicide-related stigma**<br>• "The information was given in the paper because the people and the government should know that the farmer committed suicide (*sic*)… The family was stigmatized, the stigma should be removed from the society, suicide awareness program should be conducted in the village. By doing this, others should not be stigmatized." (23-year-old male, Maharashtra, case 7) | **Family faced stigma**<br>• The deceased was accused of stealing a mobile phone. The family did not read the news themselves but were informed about it by other people. "They felt very bad and sad. It was observed that they were talking about it in the village and at work place that your son had stolen a kind of stigma." (34-year-old female, Maharashtra, case 15) |
| | **Others should not read about it and get ideas**<br>• "This news not to be published in paper so that other person will not do this type of suicide death." (55-year-old male, Maharashtra, case 16) |
| | **Felt distress on reading the news**<br>• "Family suffers and gets troubled, memories come back." (30-year-old male, Maharashtra, case 17)<br>• "Hearing or seeing it's disturbing my family members." (48-year-old female, Tamil Nadu, case 18)<br>• "We didn't like that, fearing that the life of other girl child may be affected." (25-year-old female, Tamil Nadu, case 19) |

expressed that, "suicide among younger people and women were generally more newsworthy, as they were presumed to generate greater reader empathy and curiosity" [29].

The reporting also varied by occupation of the deceased, which was related to the prevailing gendered assumptions in the Indian context about livelihood and financial responsibilities as men's domain. Male deceased farmers and salaried employees outnumbered female deceased (who engaged more in home duties) and hence received more newspaper coverage. The topic of farmers' suicides has also been determined as newsworthy by media professionals as "farmer suicides are highly politicised and attract a lot of attention," thus impacting the amount of news coverage [29]. While studies have examined the immense psychological distress faced by survivors of farmers' suicide deaths [30–33], this study captured the perceptions of newspaper coverage of those deaths on suicide loss survivors. Notably, suicide deaths that were registered with the police received over four times higher newspaper coverage than those that were not registered as police case. This could be explained by the mutually beneficial relationship between the crime

reporters and police personnel who document suicide death cases, with both relying on each other for information, verification and reporting [29].

Nearly half of the suicide loss survivors opposed newspaper coverage and one-third supported it. However, 21% refused to provide an answer and hence the findings should be interpreted within this context. This study was designed as an exploratory inquiry into how families perceive newspaper reporting of suicide deaths within their family. Given the limited prior research on this subject in India, we did not ground the analysis in existing grief, trauma, or stigma theories. Applying such frameworks at the outset risked narrowing the scope of inquiry and filtering participants' accounts through predetermined lenses. Instead, we sought to allow themes to emerge inductively from the data, providing a broad descriptive foundation. These findings can inform future research that may more directly engage with, refine, or challenge existing theoretical frameworks. Additionally, we did not seek to assess the suicide reports in the media as part of this study, hence are unable to comment on the quality of media reporting. Furthermore, we have captured survivor perceptions at one point in time given the cross-sectional nature of the study; and it may be possible that the survivor perceptions may evolve or change over time. It is also likely that other suicide survivors in the family may have different perceptions than the person who responded to our questionnaire.

Support for the newspaper coverage varied by state, with Tamil Nadu being an outlier with total lack of support. Notably, themes of support from the government, stigma, and defamation were cited both in support and in opposition to the newspaper reporting. As seen in Maharashtra, the survivors perceived the reporting to be a tool not only to eliminate stigma but also to gain access to government compensation scheme, and hence supported the newspaper coverage of their family member's death (cases 5 and 7, Table 3). In fact, the ability to claim financial compensation through the newspaper coverage of the suicide death was a major reason for support as some schemes, at the national and state level, are available in India towards compensation to the surviving kin of farmers who die by suicide [34,35]. However, survivors from Uttar Pradesh reported defamation due to reporting and that there is no point in reporting as the government will not support the family (cases 13 and 14, Table 3). Many bereaved individuals expressed opposition to media coverage, although they often could not provide specific reasons (cases 8–12, Table 3). Given the scarcity of opportunities for open discussion about suicide and its portrayal in India, these difficulties in articulating views are not unexpected but indicate potential areas for further research. Among those who provided a reason, distress was reported as one of the reasons for opposition (cases 17–19, Table 3). In a study in India previously, a media person stated that, "For us it is just impersonal information, very dry and unhuman coverage. If you go in-depth, there are a lot of complex issues at play. [But] the suicide reports are impersonal and brief" [29]. This approach to reporting style can add to the distress that suicide loss survivors reported experiencing in our study, thereby facilitating their opposition to the newspaper coverage. Indian media professionals have also described contact with the bereaved families following the suicide death as a fraught and unpleasant part of the process of reporting on suicides [29]. Also, the surviving family members face the repercussions of the loss of familial and/or economic role previously fulfilled by the deceased, in addition to having to deal with the communities within which their household is entrenched, which can complicate the bereavement process and add to the survivors' distress [36]. For example, mother of the deceased teenage boy in Maharashtra in this study reported facing gossip and discrimination at the workplace as a result of the newspaper reporting (case 15, Table 3). Another household in Tamil Nadu that was mired in poverty (case 19, Table 3), did not want news about the suicide death in their family to spread as there were young women in the family whose marriage prospects could be impacted as the result of newspaper.

Suicide loss survivors who supported newspaper coverage believed that it helped with suicide prevention (cases 1 and 2, Table 3). Menon et al claimed that suicide in the Asian context differs from those in the West due to several factors; the most important differences include a lower prevalence of mental illness among the victims and the lower men to women ratio and greater role for socio-economic determinants of suicide. Armstrong et al. [29] has documented that some journalists in India deemed coverage of suicide deaths with details important as it could highlight an injustice or impact of government policy. In this context, media may have a role in assisting suicide prevention activities in Asia by focusing on the socio-economic determinants, de-stigmatization of suicide, to move it from the domain of a 'mental illness' to more of

a 'social illness', and to encourage help seeking [16]. However, on the other hand, it is not clear how the media, at large, can report on suicide deaths to work towards suicide prevention, if the people closest to the deceased are not interested or do not like reading the news of the death as found in our study.

Implications of reading about the suicide death in newspaper varied not only by the socio-demography of the deceased but also of the survivors, highlighting the complexity of suicide bereavement, and that it needs to be understood further within the local socio-cultural context. Studies on grief interventions for suicide survivors are scarce despite the recognized negative impact of suicide stigma on the bereaved. There is a critical need for research and evidence-based recommendations on how to best to support this vulnerable population, including research combining content analysis of media reports with longitudinal interviews of survivors to capture evolving perceptions.

In conclusion, research that captures how bereaved families experience suicide reporting can play a vital role in strengthening media practice. By documenting the potential for distress, stigma, or re-traumatization, such evidence humanizes the consequences of insensitive reporting and underscores why ethical guidelines matter beyond preventing contagion. Insights from families can inform refinements to existing media codes, ensuring that they also address the needs of survivors and not only the general public. These findings could be translated into journalist training, promoting greater sensitivity in coverage and equipping reporters with practical approaches to minimize harm. Moreover, understanding circumstances in which families value media attention—for awareness or prevention—could open pathways for constructive engagement between survivors and media professionals. In this manner, research on family perspectives would help foster reporting that is both safe and respectful, balancing the public's right to information with the protection of those most impacted.

## Acknowledgments

We appreciate the contribution of informants who participated in the interviews and the field team who interviewed the respondents.

## Author contributions

**Conceptualization:** Md. Akbar, Moutushi Majumder, G. Anil Kumar, Rakhi Dandona.

**Data curation:** G. Anil Kumar, Rakhi Dandona.

**Formal analysis:** Neha Dhole, Rakhi Dandona.

**Funding acquisition:** Rakhi Dandona.

**Investigation:** Md. Akbar, Moutushi Majumder, Siva Prasad Dora.

**Methodology:** Md. Akbar, Moutushi Majumder, Siva Prasad Dora, G. Anil Kumar, Rakhi Dandona.

**Project administration:** Md. Akbar, Moutushi Majumder, Siva Prasad Dora, G. Anil Kumar.

**Software:** G. Anil Kumar.

**Supervision:** Md. Akbar, Moutushi Majumder, Siva Prasad Dora, G. Anil Kumar, Rakhi Dandona.

**Validation:** Rakhi Dandona.

**Writing – original draft:** Neha Dhole, Rakhi Dandona.

**Writing – review & editing:** G. Anil Kumar, Rakhi Dandona.

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
