## [Decision Letter · Decision Letter 0]

18 Sep 2025

PMEN-D-25-00352

Coverage of suicide deaths in newspapers and perceptions of suicide loss survivors on reporting: Insights from a community survey in India

PLOS Mental Health

Dear Dr. Dandona,

Thank you for submitting your manuscript to PLOS Mental Health. After careful consideration, we feel that it has merit but does not fully meet PLOS Mental Health’s publication criteria as it currently stands. Therefore, we invite you to submit a revised version of the manuscript that addresses the points raised during the review process.

We look forward to receiving your revised manuscript.

Kind regards,

Lambert Zixin Li, Ph.D.

Academic Editor

PLOS Mental Health

Journal Requirements:

1. Please provide a detailed online Financial Disclosure statement. This is published with the article. It must therefore be completed in full sentences and contain the exact wording you wish to be published.

a) State the initials, alongside each funding source, of each author to receive each grant, if applicable. For example: “This work was supported by the National Institutes of Health (####### to AM; ###### to CJ) and the National Science Foundation (###### to AM).”

For more information, please see our guidelines: https://journals.plos.org/digitalhealth/s/submission-guidelines#loc-financial-disclosure-statement

2. Please ensure that the funders and grant numbers match between the Financial Disclosure field and the Funding Information tab in your submission form. Note that the funders must be provided in the same order in both places as well.

3. Please update your online Competing Interests statement. If you have no competing interests to declare, please state: “The authors have declared that no competing interests exist.”

Additional Editor Comments (if provided):

Please remove all causal languages, including increase, affect, influence, and replace them with associated with, related to, or covary with to reflect your cross-sectional design. Thank you!

Reviewers' comments:

Reviewer's Responses to Questions

**Comments to the Author**

1. Does this manuscript meet PLOS Mental Health’s publication criteria?

Reviewer #1: Yes

Reviewer #2: Yes

Reviewer #3: Yes

Reviewer #4: Yes

Reviewer #5: Partly

Reviewer #6: Yes

2. Has the statistical analysis been performed appropriately and rigorously?

Reviewer #1: Yes

Reviewer #2: Yes

Reviewer #3: Yes

Reviewer #4: Yes

Reviewer #5: Yes

Reviewer #6: Yes

3. Have the authors made all data underlying the findings in their manuscript fully available (please refer to the Data Availability Statement at the start of the manuscript PDF file)?

Reviewer #1: No

Reviewer #2: Yes

Reviewer #3: Yes

Reviewer #4: No

Reviewer #5: Yes

Reviewer #6: Yes

4. Is the manuscript presented in an intelligible fashion and written in standard English?

Reviewer #1: Yes

Reviewer #2: Yes

Reviewer #3: Yes

Reviewer #4: Yes

Reviewer #5: Yes

Reviewer #6: Yes

Reviewer #1: General Comments:

This manuscript presents a timely and culturally relevant study on newspaper coverage of suicide deaths and the perceptions of suicide loss survivors in India. The topic is understudied, especially in low- and middle-income countries, and the findings have significant implications for media guidelines, postvention support, and public health policy. The methodology is generally sound, and the mixed-methods approach adds depth to the quantitative findings. However, several aspects require clarification and strengthening to enhance the manuscript's rigor and impact.

Major Comments:

Clarity on Sampling and Representativeness: The manuscript states that 155 suicide deaths were included from three states, but it is unclear how these were selected from the larger national survey of 366 suicide deaths. Please clarify the sampling strategy and justify the representativeness of the selected states and cases (e.g., are there any notable socioeconomic, cultural, or other differences between the three states selected in the article and other states).

Definition and Recruitment of “Suicide Loss Survivors”: The term “adult member most knowledgeable about the suicide” is used to define survivors. Please provide more detail on how this person was identified and whether this introduced any selection bias (e.g., gender, relationship to deceased).

Qualitative Analysis Methods: The inductive thematic analysis process is briefly mentioned but not sufficiently detailed. Please describe the coding process, how themes were derived, inter-coder reliability, and how disagreements were resolved.

Missing Demographic and Contextual Data: The socio-demographic table for survivors (Table 2) is incomplete and difficult to interpret. Please ensure all rows and columns are clearly labeled and that p-values are correctly associated with the relevant comparisons.

Theoretical Framework: The study would benefit from a stronger theoretical grounding in grief, trauma, or stigma theories to frame the findings and discussion. For instance, incorporating models of complicated grief or structural stigma could deepen the interpretation of survivors’ responses.

Implications for Practice: While the discussion touches on media guidelines, more specific recommendations for journalists, policymakers, and mental health professionals in the Indian context would strengthen the practical impact of the study.

Minor Comments:

Abstract: The abstract clearly summarizes key findings but could briefly mention the methodological approach (semi-qualitative interviews) and the three states involved.

Introduction: The introduction effectively sets the context but could better highlight the gap regarding survivors’ perspectives, which is the study’s main contribution.

Results: Some confidence intervals are very wide due to small sample sizes (e.g., female suicides in certain categories). Acknowledge this limitation when interpreting these estimates.

Tables and Figures: Figures 1 and 2 are too blurry to see their contents. High-definition images are recommended.

Language and Flow: The manuscript is well-written but would benefit from minor editing for flow and consistency (e.g., avoid repetition in the discussion).

Conclusion:

This study addresses an important gap in the literature on suicide reporting and its impact on survivors in India. With revisions to clarify methods, strengthen the theoretical framework, and expand practical implications, this manuscript will be a valuable contribution to the field of mental health and media studies.

Reviewer #2: Thank you for submitting your manuscript entitled “Coverage of suicide deaths in newspapers and perceptions of suicide loss survivors on reporting: Insights from a community survey in India” to PLOS Mental Health. I found the paper to be timely, original, and of potential high value to the field. In particular, the inclusion of survivor perspectives adds important depth to an under-researched area. However, before this manuscript can be considered further, several substantive revisions are needed.

See the attached

Reviewer #3: The article was well researched and the findings speak to issues that need to be worked on. If these findings are looked into and reviewed there could be lobbying for change in some reporting policies .As always the elite tend to be all over the media hence the high coverage of their suicide cases yet the marginalized cases are key to reform in some policies .I noted cultural beliefs ,social status also played a role in reporting suicide cases which indicates a lot was done to obtain the data.

Reviewer #4: General comments:

The paper is addressing the aspect of coverage of suicide deaths in newspapers and perceptions of suicide loss survivors on reporting following the community survey in India. However, there issues that needs to be addressed.

The authors need to define some terms such as “Suicide loss survivors” as used in the manuscript. There is also some minor grammar issues that need to be improved.

1. Introduction

The authors need to elaborate more on the types of suicide and the suicide behaviors in the introduction for clear understanding of the topic, the risk factors, perception and implication to the affected individuals and communities. The authors should also elaborate more on the World Health Organization (WHO) and the International Association for Suicide Prevention formulated guidelines.

2. Methods

The authors need to expand on the methodology. The authors should replace “Methods” with “Methodology”. In line 85, the authors state that “Deaths of all ages that occurred between 2019 and 2022….” what about those that were not reported or recorded? The authors should explain the study setting and the study population, the selection criteria, sampling procedure given that this is a sensitive topic. How were the “Suicide loss survivors” identified? And who are the “adult member” of the survivor? How was the pilot testing of the questionnaires conducted, and on how many participants and from where?

3. Results

The authors should check the numbers if they add up to the total number of the study participants as well as the 100% in the table 1. And a table for the total number and % suicide deaths for both male and female should be included in the table 1. The values in the tables indicate what? The same should be done in table 2. Generally table 1 and 2 are not very clear and the text that follow them, and they need improvement. There is need to improve on the grammar in table 3. Figure 1 and 2, need to be labeled and otherwise which is figure 1 and 2; and they need also to be improved on since they are not clear.

4. Discussion

The authors need to improve on the discussion based on their findings, and they need to link their finding with previous studies and the implication of their finding. What were the limitations of the study?

5. Conclusion

The authors need to separate the conclusion from the discussion. And the conclusions made do not seem to come from the authors’ findings i.e. results of the study.

6. References

There are some references that are incomplete such as the reference 1 and the authors need improve on the reference section.

Reviewer #5: I believe it is an interesting and valuable report on an important matter. The statistical and overall quantitative analysis appear to be well conducted and reported. However, there is an important issue about the qualitative data, as it is not explained the analysis process in the methods. For example, in the Table 3 there is a selection of quotes from the participants, but there is no information about how that selection was made. The thematic analysis should be explained in methods. This appears to me as a mixed methods paper and it should presented that way.

In the Discussion, some statements lack a more formal or scientific language, for example when they refer to a "symbiotic" relationship between journalists and police force. That should be described in better terms.

Reviewer #6: Introduction and Purpose

The study under review explores an under-researched area: how newspaper coverage of suicide deaths is perceived by suicide loss survivors in India. While global literature highlights the role of media in shaping public understanding of suicide, very little attention has been paid to how bereaved families experience such reporting. This research attempts to bridge that gap by examining both the extent of newspaper coverage and survivors’ support or opposition to it, with particular attention to socio-demographic factors.

Methodology

The researchers conducted semi-qualitative interviews with a key adult survivor from each of 155 suicide cases across three Indian states. The design is noteworthy for combining quantitative measures (e.g., proportions of cases reported in newspapers, statistical associations with socio-demographic variables) with qualitative insights (survivors’ stated reasons for support or opposition).

A strength of this design lies in its relatively large and geographically diverse sample, which enhances generalizability compared to single-site or small-sample studies. However, the reliance on one survivor per case introduces a potential bias, as perceptions may differ within families. Moreover, the “semi-qualitative” approach is not fully detailed, leaving questions about the rigor of the qualitative analysis (e.g., coding methods, inter-rater reliability).

Findings

The study reports that 39.4% of suicides were covered in newspapers, with slightly higher coverage of female suicides (43.2%) than male suicides (37.8%). Coverage was more likely when deaths were registered with the police.

Survivors’ responses to such coverage varied:

49.2% opposed it,

31.1% supported it,

19.7% declined to respond.

Support for coverage was linked to raising awareness about suicide, especially regarding farmer suicides and the broader implications of suicide. Interestingly, support for reporting male suicides rose with survivors’ education level, whereas female suicide reporting showed no clear socio-demographic patterns. Opposition was more common but often lacked articulated reasons.

Strengths of the Study

Novel focus: Few studies consider survivors’ perceptions of suicide reporting, particularly in the Indian context, where stigma, cultural norms, and media practices intersect in complex ways.

Empirical contribution: The study provides rare quantitative data on the prevalence of suicide reporting in newspapers, alongside survivors’ attitudes.

Policy relevance: Findings have clear implications for shaping media guidelines on sensitive suicide reporting.

Limitations

Analytical depth: The study’s qualitative findings are somewhat thin, as nearly half of survivors opposing coverage provided no reasons. More probing interviews or richer analysis could have deepened insights.

Contextual gaps: The study does not analyze the content or tone of newspaper reports (sensational vs. factual), which likely influences survivors’ perceptions.

Cultural considerations: While the study mentions farmer suicides, it does not fully engage with stigma, religion, or social honor, which are central to how suicide is perceived in India.

Methodological ambiguity: The “semi-qualitative” approach lacks detail, limiting replicability and transparency.

Cross-sectional design: Survivors’ views may evolve over time, but the study captures only a single point in bereavement.

Implications

The study underscores the ambivalence of survivors toward suicide reporting: while some see it as an opportunity for awareness, others experience it as invasive or stigmatizing.

This tension points to the need for:

Culturally sensitive media guidelines tailored to Indian contexts.

Training for journalists on responsible suicide reporting, balancing awareness-raising with survivor protection.

Further research combining content analysis of media reports with longitudinal interviews of survivors to capture evolving perceptions.

Conclusion

This article makes a valuable contribution by foregrounding survivors’ voices in the discourse on media and suicide. It demonstrates that the impact of newspaper reporting is neither uniformly harmful nor beneficial but varies by socio-demography, education, and cultural context. While methodological limitations constrain the depth of its conclusions, the study opens an important research agenda on suicide, media, and bereavement in India.

**Do you want your identity to be public for this peer review?** For information about this choice, including consent withdrawal, please see our Privacy Policy

Reviewer #1: No

Reviewer #2: **Yes: ** Dr David Onchonga

Reviewer #3: **Yes: ** Shepard M.M Munyoro

Reviewer #4: No

Reviewer #5: No

Reviewer #6: No

---

## [Editor Report · Decision Letter 1]

14 Oct 2025

Coverage of suicide deaths in newspapers and perceptions of suicide loss survivors on reporting: Insights from a community survey in India

PMEN-D-25-00352R1

Dear Prof Dandona,

We are pleased to inform you that your manuscript 'Coverage of suicide deaths in newspapers and perceptions of suicide loss survivors on reporting: Insights from a community survey in India' has been provisionally accepted for publication in PLOS Mental Health.

Best regards,

Lambert Zixin Li, Ph.D.

Academic Editor

PLOS Mental Health

Dear authors,

Thank you for revising your manuscript and responding carefully to the reviewers’ comments. I have reviewed your revised paper and response memo, and find the revisions satisfactory. The manuscript is now of publishable quality.

Congratulations, and thank you for choosing our journal for your work.

Kind regards,

Lambert Zixin Li